# Identification of Regulatory Molecular “Hot Spots” for LH/PLOD Collagen Glycosyltransferase Activity

**DOI:** 10.3390/ijms241311213

**Published:** 2023-07-07

**Authors:** Daiana Mattoteia, Antonella Chiapparino, Marco Fumagalli, Matteo De Marco, Francesca De Giorgi, Lisa Negro, Alberta Pinnola, Silvia Faravelli, Tony Roscioli, Luigi Scietti, Federico Forneris

**Affiliations:** 1The Armenise-Harvard Laboratory of Structural Biology, Department of Biology and Biotechnology, University of Pavia, Via Ferrata 9A, 27100 Pavia, Italy; 2NSW Health Pathology Randwick Genomics Laboratory, Prince of Wales Hospital, Sydney, NSW 2031, Australia; 3Neuroscience Research Australia (NeuRA), Prince of Wales Clinical School, University of New South Wales, Sydney, NSW 2052, Australia; 4Fondazione Istituto di Ricovero e Cura a Carattere Scientifico (IRCCS) Policlinico San Matteo, 27100 Pavia, Italy

**Keywords:** collagen biosynthesis, extracellular matrix, post-translational modifications, collagen glycosylations, glycosyltransferase, lysyl hydroxylase

## Abstract

Hydroxylysine glycosylations are post-translational modifications (PTMs) essential for the maturation and homeostasis of fibrillar and non-fibrillar collagen molecules. The multifunctional collagen lysyl hydroxylase 3 (LH3/PLOD3) and the collagen galactosyltransferase GLT25D1 are the human enzymes that have been identified as being responsible for the glycosylation of collagen lysines, although a precise description of the contribution of each enzyme to these essential PTMs has not yet been provided in the literature. LH3/PLOD3 is thought to be capable of performing two chemically distinct collagen glycosyltransferase reactions using the same catalytic site: an inverting beta-1,O-galactosylation of hydroxylysines (Gal-T) and a retaining alpha-1,2-glucosylation of galactosyl hydroxylysines (Glc-T). In this work, we have combined indirect luminescence-based assays with direct mass spectrometry-based assays and molecular structure studies to demonstrate that LH3/PLOD3 only has Glc-T activity and that GLT25D1 only has Gal-T activity. Structure-guided mutagenesis confirmed that the Glc-T activity is defined by key residues in the first-shell environment of the glycosyltransferase catalytic site as well as by long-range contributions from residues within the same glycosyltransferase (GT) domain. By solving the molecular structures and characterizing the interactions and solving the molecular structures of human LH3/PLOD3 in complex with different UDP-sugar analogs, we show how these studies could provide insights for LH3/PLOD3 glycosyltransferase inhibitor development. Collectively, our data provide new tools for the direct investigation of collagen hydroxylysine PTMs and a comprehensive overview of the complex network of shapes, charges, and interactions that enable LH3/PLOD3 glycosyltransferase activities, expanding the molecular framework and facilitating an improved understanding and manipulation of glycosyltransferase functions in biomedical applications.

## 1. Introduction

Collagens are the most abundant proteins in the human body and are highly conserved [1,2]. The different oligomeric architectures and roles of collagen molecules depend strongly on a variety of post-translational modifications (PTMs), including proline and lysine hydroxylations, as well highly specific glycosylations of hydroxylated lysines (Hyl) [1,3]. The disaccharide present in Hyl contains a highly conserved glucosyl(α-1,2)-galactosyl(β-1,O) glycan moiety, which was discovered in the late sixties [4,5,6]. Monosaccaridic galactosyl-(β-1,O)-Hyl were identified as a result of catabolic reactions carried out by the collagen α-glucosidase, an enzyme highly specific for the disaccharide substrate found on collagenous domains. The role for this enzyme is to localize collagen in the glomerular basement membrane [7,8,9].

The spread of glycosylation largely depends on collagen type [4,10,11], the functional area inside tissues [12,13,14], the developmental stage [15,16], and on disease states [17,18,19]. However, although studied extensively, the precise mechanisms of collagen glycosylation and their biological relevance in collagen homeostasis remain poorly understood.

The stereochemistry of the Glc(α-1,2)-Gal(β-1,O)-Hyl-linked carbohydrate is consistent with at least two distinct enzyme types that are required for collagen molecule PTMs [20,21]. Since both donor substrates (UDP-galactose and UDP-glucose) exhibit an α-glycosidic bond, the first reaction requires an inverting-type galactosyltransferase (Gal-T) acting on Hyl, whereas the subsequent glucosylation is catalyzed by a retaining-type glucosylgalactosyltransferase (Glc-T). Multifunctional collagen lysyl hydroxylases (LHs/PLODs) possess both Fe^2+^-dependent lysyl hydroxylase (LH) and Mn^2+^-dependent glucosyltransferase activities [22,23,24]. The human isoenzyme 3 (LH3/PLOD3) was reported to also exhibit Gal-T activity in vitro [22]. In vivo studies have demonstrated that decreased LH3/PLOD3 protein levels and/or pathogenic mutations in the glycosyltransferase (GT) domain exclusively impair Glc-T activity [18,25,26]. This occurs secondarily to the LH3/PLOD3 p.(Asn223Ser), which introduces an additional glycosylation site within the enzyme’s GT domain, leading to osteogenesis imperfecta [18], and in the recently identified LH3/PLOD3 p.(Pro270Leu), which results in a Stickler-like syndrome with vascular complications and variable features typical of Ehlers–Danlos syndromes and Epidermolysis Bullosa [26]. Mouse studies have also shown that only the LH3/PLOD3 Glc-T activity is indispensable for the biosynthesis of collagen IV and formation in the basement membrane during embryonic development [15,27], consistent with the presence of additional collagen galactosyltransferases. To date, two genes encoding for Mn^2+^-dependent *O*-galactosyltransferases (GLT25D1 and GLT25D2) have been identified [28,29], and GLT25D1 has been proposed to act in concert with LH3/PLOD3 on collagen molecules [28,30,31]. Studies on osteosarcoma cell lines, which produce large amounts of fibrillar collagens, have shown that the simultaneous deletion of GLT25D1 and GLT25D2 results in growth arrest due to a lack of glycosylation, causing some to hypothesize that the Gal-T activity of LH3/PLOD3 might not be as essential as the Glc-T activity [32].

These data support the hypothesis that, in vivo, the entire collagen glycosylation machinery may involve distinct proteins and protein complexes for Gal-T and Glc-T reactions. This raises the intriguing question of how this highly conserved process is spatiotemporally regulated at the molecular level. However, our current understanding of collagen glycosyltransferases is restricted to three-dimensional structures of human LH3/PLOD3 in complex with UDP-sugar donor substrates [33] and the few mutagenesis studies focusing on the hallmarks of Mn^2+^-dependent glycosyltransferase catalysis [22,34].

By combining structure-guided mutagenesis with conventional indirect activity assays and a new mass spectrometry-based direct assay to monitor the Lys-to-Glc-Gal-Hyl conversion altogether, we identified a broad group of amino acid residues in the LH3/PLOD3 GT domain that are cooperatively responsible for its glycosyltransferase activity. The remarkable conservation of the amino acid network, which shapes the GT domain in all human LH/PLOD isoforms, combined with recent reports of Glc-T activity for LH1/PLOD1 [23] and LH2/PLOD2 [24] extends the significance of the proposed network, providing a detailed molecular blueprint of the Glc-T activity hallmarks belonging to the entire enzyme family. The results obtained in this study on GLT25D1 illustrate that this enzyme is the sole responsible for Gal-T activity on Hyl residues and rule out the role of LH3/PLOD3. GLT25D1 generates the substrate for the LH/PLOD-mediated Glc-T reaction, ultimately yielding Glc-Gal-Hyl. Finally, we also identified and characterized UDP-sugar substrate analogs acting as inhibitors of LH3/PLOD3 Glc-T activity, providing templates for the development of the first collagen Glc-T inhibitors.

## 2. Results

### 2.1. A Direct MS-Based Assay to Evaluate Lys-to-Glc-Gal-Hyl Conversion

The results from previous studies have consistently highlighted that the N-terminal GT domain is the only catalytic glycosyltransferase domain of LH/PLOD enzymes [21,23,24,33]. In this respect, the question of the controversial possible coexistence of two chemically opposite glycosyltransferase catalytic activities (i.e., the inverting Gal-T and the retaining Glc-T activities) within the same catalytic site is still open. The recent identification of the collagen galactosyltransferase family GLT25D1/2 has provoked questions about the initially observed LH3/PLOD3 Gal-T reactivity [33]. Therefore, we recombinantly produced both human LH3/PLOD3 and GLT25D1 and optimized the previously established MS-based assay used to monitor Fe^2+^-dependent Lys-to-Hyl conversion [33] to directly detect the presence of Gal-Hyl and Glc-Gal-Hyl residues on synthetic collagen peptides (Figure 1A). Firstly, the LH activity of LH3/PLOD3 was tested for both enzymes in the presence of all of the necessary cofactors (Fe^2+^, 2-OG and ascorbate) and the acceptor substrate (i.e., synthetic collagen peptide (GIK)_4_), observing the expected introduction of a hydroxyl group (16 Da) on a single Lys residue (Figure 1B and Appendix A). The same reaction mixture containing GLT25D1 and not LH3/PLOD3 did not produce any modifications to the substrate peptides (Appendix A), supporting the idea that GLT25D1 exclusively accepts Hyl residues as substrates (Figure 1A). To assess the putative inverting Gal-T activity of LH3/PLOD3, Mn^2+^ and UDP-Gal (i.e., cofactors of GalT reaction) were added to the reaction mixture. Surprisingly, no additional peaks were observed (Figure 1C). By replicating the same assay with the addition of GLT25D1, the MS spectrum revealed an additional peak compatible with the galactosylation of a single Hyl residue (Figure 1D). These results unambiguously establish that GLT25D1 (not LH3/PLOD3) is solely responsible for the galactosylation of collagen Hyl residues. Finally, the incorporation of UDP-Glc into the reaction mixture resulted in the occurrence of a fourth MS peak, corresponding to the expected mass of the nude peptide with a single Glc-Gal-Hyl residue (Figure 1E).

### 2.2. The Amino Acid Residues Shaping the UDP Binding Site Are Essential for LH3/PLOD3 Glc-T Activity

The LH3/PLOD3 N-terminal GT domain shares its fold with Mn^2+^-dependent GT-A glycosyltransferases, encompassing a UDP-donor substrate binding cavity stretched towards a catalytic pocket [33]. To date, the Glc-T activity of homologous LH1/PLOD1 and LH2/PLOD2 enzymes has been poorly characterized. Using indirect luminescence-based assays, we previously reported the LH1/PLOD1 Glc-T activity to be ten-fold less than LH3/PLOD3 [23]. Other studies also reported on the possible role of the LH2/PLOD2 Glc-T activity in lung adenocarcinoma progression [24]. To identify key residues in the catalytic site and dissect their contribution to the LH3/PLOD3 Glc-T activity, we used previously generated homology models of LH1/PLOD1 and LH2/PLOD2 [35] and performed a comparative structural analysis between the GT domains of all human LH/PLOD variants (Appendix A). Overall, the models for LH1 and LH2 could superimpose well with the crystal structure of LH3/PLOD3, enabling an accurate comparison at the amino acid side chain level (Appendix A).

We inspected the UDP-binding cavity to identify distinguishing features by comparing the amino acids found in LH3/PLOD3 with those predicted for the LH1/PLOD1 and LH2/PLOD2 isoforms, which were described to have low Glc-T activity [23,24]. We observed that nearly all of the residues involved in Mn^2+^ and UDP binding are conserved in all paralogs, except for Val80, becoming Lys68 in LH1/PLOD1 and Gly80 in LH2/PLOD2 (Figure 2A and Appendix A). The presence of a different amino acid side chain surrounding the donor substrate cavity led us to consider whether this could be a discriminating functional feature among the GT domains in LH/PLOD enzymes. In LH3/PLOD3, Val80 is located in the middle of a flexible “glycoloop” (Gly72-Gly87), not visible in the electron density of the ligand-free LH3/PLOD3 structure, and stabilized upon UDP-substrate binding [33]. Within the glycoloop, residue Val80 is in close proximity to the ribose ring of the UDP-sugar donor substrates. We hypothesized that the introduction of a large, positively charged residue such as Lys in LH1/PLOD1 or alterations due to the complete removal of side chain steric hindrance such as Gly in LH2/PLOD2 could interfere with the binding of donor substrates. Therefore, we generated the LH3/PLOD3 p.(Val80Lys) and p.(Val80Gly) mutants. These enzyme variants were found to be folded based on analytical gel filtration and differential scanning fluorimetry (DSF) and showed lysine hydroxylation activity comparable to wild-type LH3/PLOD3 (Appendix A). Conversely, both mutations resulted in non-detectable Glc-T activity in MS-based assays (Figure 2B, Table 1). Moreover, since LH3/PLOD3 is capable of activating donor UDP-sugar substrates and releases UDP in the absence of the acceptor collagen substrate [33], we investigated the impact of these mutations in both the absence (uncoupled activity) and presence (coupled activity) of acceptor substrates by performing luminescence-based assays, obtaining very similar results using both enzyme variants. These experiments demonstrate that the Glc-T activity of LH3/PLOD3 p.(Val80Lys) and p.(Val80Gly) is mostly imputable to the uncoupled activity (Figure 2C, Table 1). Consistently, Val80 might be involved in the productive positioning of the donor substrate to enable the transfer of the glycan moiety to the collagen acceptor substrate rather than stabilizing the UDP moiety in the catalytic pocket.

To further investigate the influence of LH3/PLOD3 Val80 on Glc-T activity, we crystallized and solved the 3.0-Å resolution structure of the p.(Val80Lys) mutant in complex with Mn^2+^ and also obtained the 2.3-Å resolution structure of the same mutant in the presence of both Mn^2+^ and the UDP-Glc donor substrate (Figure 3A, Appendix A). Overall, both structures superimpose almost perfectly with wild-type LH3/PLOD3 for all domains (Appendix A). The structure of the LH3/PLOD3 p.(Val80Lys) mutant bound to Mn^2+^ is essentially identical to that of wild-type LH3/PLOD3 in complex with the same cofactors. In both structures, the glycoloop containing Val/Lys80 could not be modelled in the experimental electron density due to its high flexibility (Figure 3C). On the other hand, the side chain of the Lys80 residue could be modelled unambiguously in the experimental electron density of the UDP-donor substrate-bound structure. Despite the increased steric hindrance, the mutated Lys80 residue adopted a conformation compatible with the simultaneous presence of the UDP-Glc in the catalytic cavity. However, similar to what was observed for wild-type LH3/PLOD3, the glycan moiety of UDP-Glc was not visible in the electron density (Figure 3A). Collectively, these data are consistent with the alteration introduced by the p.(Val80Lys) mutation impacting on the LH3/PLOD3 glycosyltransferase catalytic activities.

### 2.3. The LH3/PLOD3 Glc-T Activity Is Affected by the Long-Range Rearrangement of Trp92 and Trp75

The glycoloop is a structural feature found exclusively in the GT domains of LH/PLOD enzymes. It incorporates Trp75, a residue whose aromatic side chain stabilizes the uridine moiety of the donor substrate through pi-stacking and, together with residue Tyr114 of the DxxD motif (a distinguishing feature of LH/PLOD GT domain, [33]), “sandwiches” the donor substrate in an aromatic stacking environment (Figure 2A). Both residues are critical for LH3/PLOD3 Glc-T enzymatic activity [33]. However, the conformation adopted by the LH3/PLOD3 glycoloop in the presence of UDP-donor substrates is not accompanied by other significant structural changes in the surrounding amino acids, except for the minor rearrangements of distant residue Trp92—which is not conserved in other LH/PLOD isoenzymes (Figure 2A and Appendix A)—whose bulky side chain rearranges, pointing towards the aromatic ring of Trp75. Prompted by this observation, we mutated this residue to alanine and found that the presence of this variant did not alter the folding of the enzyme nor its LH/PLOD enzymatic activity (Appendix A). Conversely, the p.(Trp75Ala) mutant Glc-T activity is undetectable compared to wild-type LH3/PLOD3 (Figure 2B,C, Table 1), and the impact of the p.(Trp92Ala) mutation seemed to slightly affect Glc-T activity (Figure 2B,C, Table 1). These findings suggest that the presence of non-conserved residues within the LH3/PLOD3 GT domain that are distant from those involved in first-shell interactions with the donor and acceptor substrates may contribute to the productive conformations of the glycoloop in donor substrate-bound states.

### 2.4. A Poly-Asp Sequence near the Donor Sugar Binding Site Is Essential for Glc-T Activity in LH1/PLOD1 and in LH3/PLOD3

The poly-Asp sequence (Asp188-Asp191, Figure 2A) is visible in UDP-sugar-bound structures and in contact with the glycoloop. This sequence is partially conserved in LH/PLOD isoforms (Appendix A), and mutations of Asp190 and Asp191 have also been reported to affect the glycosyltransferase enzymatic activities of LH3/PLOD3 [34]. Based on LH3/PLOD3 crystal structures, such behavior is expected since residues Asp190 and Asp191 point towards the Glc-T catalytic cavity. We designed and generated individual alanine mutants for both LH3/PLOD3 Asp190 and Asp191 that showed that both variants were compatible with folded and functional LH/PLOD enzymes (Appendix A). When tested for Glc-T activity, both these mutants resulted inactive, as confirmed by MS and the luminescence-based assay results (Figure 2B,C, Table 1). Collectively, these data are consistent with the involvement of the residues of the poly-Asp repeat, and in particular Asp190 and Asp191, in both the positioning and recognition of donor or acceptor substrates.

A recent report showed that the p.(Ser178Arg) mutation on LH1/PLOD1 (corresponding to LH3/PLOD3 Asp190 in the poly-Asp sequence) caused a heritable thoracic aortic disease [23]. The p.(Asp190Arg) mutation strongly affected the LH3/PLOD3 enzyme function by causing a drastic drop in Glc-T catalytic activity. Aiming to further elucidate the significance of the differences caused by this specific residue in LH1/PLOD1 and LH3/PLOD3, we generated a LH3/PLOD3 p.(Asp190Ser) variant to mimic LH1/PLOD1 at that specific site (Appendix A). MS and luminescence data showed that this single-point mutation completely abolishes the LH3/PLOD3 Glc-T activity (Figure 2B,C) without affecting the ability of this mutant to bind UDP-Glc donor substrates (Appendix A). We also solved the 2.30-Å crystal structure of the full-length human LH3/PLOD3 p.(Asp190Ser) mutant in complex with Mn^2+^ and UDP-Glc. Overall, the structure appeared nearly identical to wild-type LH3/PLOD3 (Appendix A). Analysis of the electron density at the GT catalytic site showed clear signals for the Glc moiety of the UDP-Glc donor substrate (Figure 3B). This feature was never observed when using wild-type LH3/PLOD3 [33], further supporting the importance of the Poly-Asp loop for Glc-T activity. The side chains of Asp191, Lys89, and Ser166, as well as the main chain carbonyl of Thr83, were found at distances and orientations possibly compatible with electrostatic contacts with the Glc moiety of the donor substrate, resulting in the trapping of the sugar in a portion of the catalytic cavity, leaving a large pocket shaped by residues Glu141, Phe143, Asn165, Asn223, and Asn255 available to likely host the Gal moiety of the acceptor substrate for catalysis (Figure 4).

To confirm the importance of this residue, we also investigated how this site could affect the Glc-T activity of the human LH1/PLOD1 counterpart by investigating a LH1/PLOD1 p.(Ser178Asp) mutant. Strikingly, this substitution resulted significantly rescued the LH1/PLOD1 Glc-T activity, which could be detected using both direct and indirect assays (Figure 5). Together, these data provide experimental evidence for in vitro LH1/PLOD1 Glc-T activity and further confirm the critical role for Asp190 in the activation of the donor substrate prior to its transfer to the collagen acceptor substrate.

### 2.5. Two Gating Trp Residues Modulate Glc-T Activity by Affecting Acceptor Substrate Binding

After investigating the amino acid residues involved in stabilizing the UDP moiety of donor substrates, we focused on another group of residues within the GT catalytic pocket opposite to the putative position of the flexible sugar rings of the same substrates (Figure 2A). Many LH3/PLOD3 residues shaping this part of the glycosyltransferase catalytic cavity matched the catalytic amino acids found in other GT-A type glycosyltransferases (Appendix A) [37,38]. In particular, LH3/PLOD3 Trp145, a residue located in one of the loops of the GT domain uniquely found in LH3/PLOD3, was previously suggested as a possible candidate for the modulation of Glc-T activity. This residue was found to adopt different side chain conformations in substrate-free and substrate-bound structures, affecting the shape and steric hindrance of the enzyme’s catalytic cavity [33]. Interestingly, nearly identical conformational changes were also observed when comparing substrate-free and substrate-bound LH3/PLOD3 p.(Val80Lys) and p.(Asp190Ser) structures (Figure 3C and Figure 4A). Mutating Trp145 residue into alanine completely abolishes Glc-T enzymatic activity (Figure 2B,C) without affecting protein folding stability and LH activity (Appendix A). This supports previous the hypotheses of a “gating” role for Trp145 in GT catalytic cavity, assisting the productive positioning of sugar moieties of donor substrates for effective transfer during catalysis and/or shaping the pocket to host the Lys-O-Gal moiety of the collagen acceptor substrate [33]. A comparison with molecular structures of other glycosyltransferases (including distant homologs) highlighted that most structurally related enzymes position aromatic side chains from different structural elements of their fold in their catalytic cavities in a structural arrangement that is similar to that of Trp145 in LH3/PLOD3. In particular, similar aromatic residues were found in other glycosyltransferases, such as Tyr186 in LgtC from Neisseria meningitidis, Trp314 in the N-acetyllactosaminide α-1,3-galactosyl transferase GGTA1, Trp300 in the histo-blood group ABO system transferase, and Trp243 and Phe245 in the two glucoronyltransferases B3GAT3 and B3GAT1, respectively (Appendix A, Appendix A). This is surprising considering that the loop encompassing residues 142–163 is a unique structural feature of LH/PLOD enzymes [33]. This further highlights the high versatility of glycosyltransferases, displaying an impressive structural plasticity to carry out reactions characterized by a very similar mechanism on a large variety of specific donor and acceptor substrates.

Our previous structural comparisons of ligand-free and substrate-bound LH3/PLOD3 suggested the intriguing additional possibility of a concerted mechanism involving conformational changes in a non-conserved aromatic residue located on the enzyme’s surface (Trp148, Figure 2A) together with Trp145 [33]. To investigate such a possibility, we mutated Trp148 into alanine. The resulting mutant enzyme was folded and displayed LH activity comparable to wild-type LH3/PLOD3 (Appendix A). However, glycosyltransferase assays showed that this variant had reduced Glc-T activity compared to the wild-type in the presence of both donor and acceptor substrates (Figure 2B,C, Table 1). Despite the less pronounced alterations than those observed when mutating Trp145, these data allow us to speculate about possible synergistic mechanisms between long-range substrate recognition on the enzyme’s surface and conformational rearrangements upon substrate binding in the enzyme’s catalytic site.

### 2.6. Additional Residues Facing Both Donor and Acceptor Substrates Affect the LH3/PLOD3 Glc-T Activity

The molecular structures of LH3/PLOD3 in complex with UDP-Glc and Mn^2+^ showed very weak electron density near the UDP pyrophosphate group, likely representative of multiple conformations simultaneously trapped in the substrate binding cavity [33]. We explored the LH3/PLOD3 catalytic cavity in its proximity, looking for additional amino acids potentially critical for catalysis. In particular, we searched for residues carrying carboxylic or amide side chains, which would mean they were capable of acting as candidate catalytic nucleophiles for the formation of a (covalent) glycosyl-enzyme intermediate prior to the glycosylation of the acceptor substrate [39].

In retaining-type glycosyltransferases belonging to the GT-6 family, a conserved glutamate has been found to act as a nucleophile [40,41,42]. In LH3/PLOD3 structures, we noticed that, despite being distant from the donor substrate, residues Gln192, Asn165, and Glu141 point towards the cavity that accommodates the glycan moiety (Figure 2A). We generated Ala mutations of all these residues, obtaining folded functional LH enzymes in all cases (Appendix A). When probed for Glc-T activity, we found that both p.(Asn165Ala) and p.(Gln192Ala) mutants were still capable of activating UDP-donor substrates, albeit less efficiently (Figure 2C, Table 1); however, no sugar transfer to acceptor substrates could be detected (Figure 2B, Table 1). Conversely, the p.(Glu141Ala) mutant was completely inactive and incapable of UDP-donor substrate activation (Figure 2B,C, Table 1). These data suggest essential roles for Glu141 in catalysis, either through the initial binding of UDP-Glc donor substrates prior to their final positioning in the catalytic site or through the stabilization of water molecule networks in the large GT cavity, enabling donor substrate processing. The surrounding negatively charged pocket composed of Asp190, Asp191, Gln192, Asn165, and all residues found relevant to (but not essential for) catalysis likely aids glycosyltransferase activity. Proximate to Glu141 in the GlcT cavity, residue Asn255 is the closest amino acid to the UDP phosphate–sugar bond. Despite being fully conserved in LH/PLOD isoforms (Appendix A), to date, it has not been found in any GT-A-type glycosyltransferases with known structures. The side chain of Asn255 consistently points to a direction opposite the donor substrate in all LH3/PLOD3 structures (Figure 2A). We hypothesize that the side chain amide group might also be involved in catalysis, possibly through the recognition of collagen acceptor substrates given the conformation displayed by this side chain. However, the LH3/PLOD3 p.(Asn255Ala) mutant folded properly (Appendix A) and showed that Glc-T enzymatic activity was only slightly reduced (Figure 2B,C, Table 1).

### 2.7. Pathogenic Mutations in the LH3/PLOD3 GT Domain Affect Protein Folding

Recently a pathogenic LH3/PLOD3 mutation, p.(Pro270Leu), has been identified and mapped at the interface of the AC and GT domains [26]. This proline residue localizes in a loop responsible for shaping the GT cavity; however, given its position, it is unlikely to play direct roles in catalysis. To better understand the impact of this pathogenic mutation on LH3/PLOD3 enzymatic activity, we attempted the recombinant production of a p.(Pro270Leu) LH3/PLOD3 mutant. In this case, due to the almost non-detectable protein levels, we could not reliably carry out any in vitro investigations. Considering the high reproducibility associated with the recombinant production of a large variety of LH3/PLOD3 point mutants, this result may indicate that this mutation is likely to severely impact the overall enzyme stability rather than its enzymatic activity, resulting in extremely low protein expression levels in vitro and, most likely, in vivo as well.

### 2.8. The Molecular Structures of LH3/PLOD3 in Complex with UDP-Sugar Analogs Provide Insights into the Processing of Glycan Moieties in the Catalytic Cavity

A frequent limitation associated with molecular characterizations of glycosyltransferases is the high flexibility of the donor substrate glycan moiety within the catalytic cavity. Such a limitation becomes even more relevant when the enzyme is capable of processing UDP-sugar molecules in the absence of acceptor substrates, such as in the case of LH3/PLOD3. Considering our previous [33] and current co-crystallization results, we considered whether free UDP, the product of the enzymatic reaction, could remain bound in the LH3/PLOD3 GT domain with the same efficiency as physiological donor substrates even after processing. Therefore, we compared the binding of free UDP and donor UDP-sugars using DSF and detected a thermal shift of 3–3.5 °C for free UDP compared to a 2–2.5 °C shift using UDP-sugar substrates (Figure 6A). These results were consistent with free UDP binding to LH3/PLOD3, likely with an even higher affinity than the UDP-glycan substrates, indicating that the Glc-T reaction may therefore be affected by product inhibition. Surprisingly, the increase in thermal stability did not correlate with the efficient trapping of the reaction product in wild-type LH3/PLOD3 molecular structures. Independently of the UDP concentration used in co-crystallization and soaking experiments, electron density for free UDP was not observed, yielding molecular structures completely identical to ligand-free enzymes (Appendix A).

A report by Kivirikko and colleagues [43] suggested that UDP-glucuronic acid (UDP-GlcA) could act as a competitive inhibitor of collagen glycosyltransferases. Based on this, UDP-GlcA was used to isolate LH3/PLOD3 from chicken embryo preparation [22,44]. However, no follow-up biochemical studies were published. We used DSF and luminescence-based Glc-T activity assays to investigate how UDP-GlcA could affect LH3/PLOD3 enzymatic activity. DSF showed that UDP-GlcA indeed binds weakly, resulting in a thermal shift of 1–1.5 °C (Figure 6A), highlighting limited stabilization compared to the UDP-glycan substrates and free UDP. Enzymatic assays also confirmed the weak competitive inhibition displayed by this molecule (Figure 6B), with IC_50_ values in the millimolar range (Table 2). We also successfully co-crystallized and determined the 2.2-Å resolution crystal structure of wild-type LH3/PLOD3 in complex with Mn^2+^ and UDP-GlcA (Appendix A) and found that the inhibitor could efficiently replace UDP-sugar donor substrates in the substrate cavity (Figure 6C). We observed additional experimental electron density for the glucuronic acid moiety of the inhibitor in the enzyme’s catalytic cavity; however, this density could not be interpreted with a single inhibitor conformation. Nevertheless, it unambiguously showed that the sugar adopts a bent conformation resembling that observed for UDP-Glc in the LH3/PLOD3 p.(Asp190Ser) mutant crystal structure. The glycan moiety is deeply buried in the enzyme’s catalytic cavity proximate to residues Lys89, Asp190, Asp191 but distant from the residues found critical for catalysis, including Trp145, Asn255, and Glu141 (Figure 6C), thereby leaving the remaining space in the cavity to accommodate acceptor substrates.

Considering the possible conformations adopted by the GlcA based on analysis of the electron density and the close proximity of the glucuronic acid moiety to LH3/PLOD3 Val80, we wondered whether the p.(Val80Lys) mutation could interfere with inhibitor binding. Therefore, we co-crystallized and solved the 2.7-Å resolution structure of the LH3/PLOD3 p.(Val80Lys) mutant in complex with UDP-GlcA (Appendix A) and surprisingly observed the partial displacement of the glycoloop, for which we could not observe the typical well-defined electron density present in UDP-sugar-bound wild-type LH3/PLOD3 structures (Figure 6D). At the same time, we could not observe improvements in the quality of the electron density for the glucuronic acid moiety, which was worse compared to that observed in wild-type LH3/PLOD3. This is consistent with the intrinsic flexibility of the sugar-like moiety not being influenced by specific conformations of the glycoloop, but rather by the lack of specific protein–ligand interactions that could stabilize the sugar ring in a unique structural arrangement.

The absence of a precise conformation for the glucuronic acid moiety observed in crystal structures prompted us to further investigate another UDP-sugar substrate analog, UDP-Xylose (UDP-Xyl), which is characterized by a lack of the carboxylic moiety of UDP-GlcA. Similar to UDP-GlcA, UDP-Xyl was shown to be a weak inhibitor of LH3/PLOD3 Glc-T activity (Figure 6A,B), with IC50 in the high micromolar range (Table 2). The 2.0-Å resolution structure of LH3/PLOD3 in complex with Mn^2+^ and UDP-Xyl also showed that the inhibitor was bound inside the enzyme’s catalytic cavity, with weak electron density associated with the sugar moiety, suggesting multiple conformations of the xylose moiety attached to UDP, similar to what was observed for UDP-GlcA (Figure 6E). Taken together, these results suggest that the LH3/PLOD3 GT cavity can host a variety of UDP-sugar substrates and that inhibition likely depends on the reduced flexibility (and therefore increased stabilization) of the ligand within the cavity. In this respect, it may be expected that UDP-sugar analogs strongly interacting with side chains proximate to the glycan moieties of UDP-GlcA and UDP-Xyl may have the potential to become powerful inhibitors of LH/PLOD glycosyltransferase activity.

## 3. Discussion

Glycosyltransferases are highly versatile enzymes with broad substrate specificity. When investigated carefully, they show a series of recurrent features that allow their comparative characterization even in the presence of low sequence/structure conservation. LH3/PLOD3 is widely known to be a multifunctional enzyme that is able to catalyze multiple steps of the Lys-to-Glc-Gal-Hyl PTM pathway [1]. Recent reports have also suggested similar multifunctional activities for the paralogs LH1/PLOD1 and LH2/PLOD2 [23,24], but direct evidence for the final Glc-Gal-Hyl product generated is still missing. Our in vitro investigations combining direct MS-based assays with the indirect detection of glucosyltransferase activity products shed light on the specific functions of this unique enzyme family. In this work, for the first time, it has been shown that LH/PLOD enzymes are exclusively involved in the glucosylation of galactosyl hydroxylysines, whereas collagen galactosyltransferases, such as GLT25D1, are solely responsible for Hyl galactosylation. Thus, our results confirm that LH/PLOD enzymes are retaining-type glucosyltransferases. Furthermore, the direct MS-based assay allowed us to detect the formation of the first fully synthetic Glc-Gal-Hyl collagen peptide in vitro through one-pot total synthesis.

A structure-guided mutagenesis investigation was performed to fully investigate the LH/PLOD enzyme’s GT domains by focusing on only non-matching residue present within the amino acid sequence directly surrounding the UDP portion of the donor substrate. This residue (Val80 in LH3/PLOD3, which corresponds to Lys68 in LH1/PLOD1 and Gly80 in LH2/PLOD2) is located in the middle of the glycoloop, in close proximity to the ribose ring of the UDP-sugar donor substrate(s). The biochemical data described here are consistent with the p.(Val80Lys) and p.(Val80Gly) substitutions impairing the LH3/PLOD3 catalytic activity. This result was corroborated by the disorder observed around the glycoloop of the LH3/PLOD3 p.(Val80Lys) crystal structure, thus providing a mechanistic explanation for the lack of Glc-T activity. It is likely that Val80 assists the positioning of the glycan moiety of the bound donor substrate. Alternatively, Val80 might be involved in the productive positioning of the donor substrate during the transfer of the glycan moiety to the collagen acceptor substrate.

By comparing the apo and UDP-Glc-bound LH3/PLOD3, we identified three residues in the second-shell environment that were adopting different conformations at the side chain level, somehow suggesting a possible involvement in the catalytic mechanism. The non-conserved LH3/PLOD3 Trp92 (Leu80 in LH1, Leu92 in LH2a/b) positions its aromatic side chain in a conformation that stabilizes the entire glycoloop to facilitate enzymatic reactions. However, the mutation of this residue did not lead to a full loss of Glc-T activity, suggesting that the folding topology of the LH/PLOD GT domain is highly versatile and adaptable to multiple amino acid alterations without abolishing Glc-T catalytic activity.

The sequence discrepancy in another critical site of the GT domain, the poly-Asp helix, which is known to be critical for catalysis, was explored [34]. It was found that LH3/PLOD3 Asp190, positioned at the entrance of the catalytic site, is critical for enzymatic activity. LH3/PLOD3 p.(Asp190Ser), matching the corresponding residue in LH1/PLOD1, resulted in strongly diminished Glc-T activity and yielded a crystal structure with observable electron density for the donor sugar moiety for the first time, enabling the identification of the amino acid residues directly involved in Glc binding and those shaping the portion of the cavity likely hosting the Gal moiety of the acceptor substrate. Strikingly, the reverse mutation in LH1/PLOD1 (Ser178Asp) strongly enhanced the Glc-T activity compared to wild-type, confirming this to be a key residue essential for the catalytic processing of the donor glucose prior to its transfer to the acceptor substrate.

At the entrance of the GT catalytic pocket, LH3/PLOD3 exhibits a non-conserved loop shaping the GT catalytic cavity bearing two aromatic residues: the Trp145 and the Trp148, which seem to act in a concerted way during catalysis, as suggested by comparisons between substrate-free and substrate-bound molecular structures (Figure 2A). Trp145 is indispensable for Glc-T activity; its conformational changes consistently respond to the presence and positioning of the donor substrate inside the catalytic cavity (Figure 3C and Figure 4). Although located in a loop that is uniquely found in LH3/PLOD3, the Trp145 side chain matches a site frequently occupied by bulky aromatic residues in other GT-A glycosyltransferases that shape a portion of the GT cavity to facilitate donor substrate processing and catalysis. This further highlights the versatility of glycosyltransferases, in which many different structural features have evolved to specifically recognize distinct donor and acceptor substrates while preserving the ability to carry out the same catalytic reaction. The implication in catalysis of the less conserved Trp148 on the surface of the GT domain is less pronounced. This residue might respond to the rearrangements of its counterpart Trp145 in the catalytic site or contribute to long-range stabilizing interactions with collagen molecules while they dock their Hyl or Gal-Hyl residues in the acceptor substrate site during the Glc-T reaction, respectively.

Catalytic nucleophiles have been clearly identified so far only in the retaining-type glycosyltransferases belonging to the GT-6 family [41,42], such as the α-1,3 galactosyltransferase (GGTA1), where a conserved glutamate is found positioned on the β-face of the donor sugar [40,45,46] (Appendix A). Conversely, extensive structural comparisons and mutagenesis experiments focusing on matching residue Gln189 have been performed in the *O*-galactosyltransferase LgtC from *Neisseria menengitidis* [47]. However, the location of this residue as a catalytic nucleophile in LH3/PLOD3 is inconsistent with its proposed function, as this site is occupied by Gln192. Such a residue is next to the poly-Asp helix, distant from the sites occupied by donor substrates and in an arrangement that is not compatible with a direct role in catalysis. Mutagenesis results indicate that the removal of the Gln192 side chain has a strong impact on LH3/PLOD3 glycosyltransferase activity; however, indirect assays show that the enzyme is still partially active (Figure 2C). Nearby, we identified two other amino acid residues potentially involved in donor substrate activation or the transfer of sugar moieties to the acceptor molecule. Both Glu141 and Asn165 point directly towards the glycan moiety of the donor substrate (Figure 2A). Whilst the Asn165Ala was incapable of Glc transfer to acceptor substrates but still capable of UDP-Glc uncoupling (Figure 2B,C, Table 1), we found that the carboxylate moiety of Glu141 is essential for catalysis, as the Glu141Ala mutation yields a completely inactive LH3/PLOD3 glycosyltransferase. Glu141 adopts a conformation corresponding to Asp130 in the *O*-galactosyltransferase LgtC from *Neisseria meningitidis*, Asp125 in the *O*-glucosyltransferase GYG1 from rabbit, and Gln247 in the *O*-glucosyltransferase GGTA1 from *Bos taurus* (Appendix A, Appendix A). As the removal of this carboxylate side chain completely abolishes LH3/PLOD3 glycosyltransferase activities, we suggest that the site corresponding to Glu141 might be a critical “hot spot” during catalysis in the glycosyltransferases characterized by the carboxylate same or amide side chains in similar positions.

Our work provides a comprehensive update to the biochemical landscape for the characterization of the Lys-to-Glc-Gal-Hyl conversion pathway based on reliable orthogonal in vitro assays and a new set of three-dimensional structures of LH3/PLOD3 and its mutants. The absence of experimental density in the co-crystal structures of LH3/PLOD3 with free UDP indicates that the retention of donor substrates in the GT domain of LH/PLODs requires the presence of both the UDP and the sugar moiety and that, regardless of the presence of glycan acceptor substrates, after hydrolysis, free UDP is rapidly released from the catalytic site, possibly through conformational changes involving the flexible glycoloop. In this respect, despite the high flexibility of the glycan moieties of the bound molecules, the new data obtained for LH3/PLOD3 in complex with UDP-sugar analogs acting as mild inhibitors (Figure 6) provide valuable insights for the structure-based drug development LH3/PLOD3 Glc-T enzymatic activity inhibitors. Whilst co-crystal structures with donor substrate consistently display no density for the sugar moiety because of uncoupled enzymatic hydrolysis or very low occupancy for a very flexible moiety, the inhibitors show density for the glycan part connected to UDP because hydrolysis is not taking place. This scenario is very similar to what we observed for the LH3/PLOD3 p.(Asp190Ser) mutant, where a significant reduction in catalytic activity (with no alterations in UDP-Glc binding affinity) results in a visible Glc moiety in the catalytic cavity (Figure 3B and Figure 4). Therefore, the insights obtained using UDP-GlcA and UDP-Xyl molecules may offer opportunities for the development of innovative therapeutic strategies against pathological conditions characterized by excess collagen glycosylations, such as osteogenesis imperfecta [48]. Together with the mutagenesis scanning of the entire GT catalytic site, our work provides a comprehensive overview of the complex network of shapes, charges, and interactions that enable LH3/PLOD3 Glc-T activities.

## 4. Materials and Methods

### 4.1. Chemicals

All chemicals were purchased from Sigma-Aldrich (Merck) (St. Louis, MO, USA), unless otherwise specified.

### 4.2. Molecular Cloning and Site-Directed Mutagenesis

The LH3/PLOD3 (UniProt Q60568—obtained from Source Bioscience, Nottingham, UK), the LH1/PLOD1 (UniProt Q02809—obtained from Source Bioscience), and the GLT25D1 (UniProt Q8NBJ5—obtained as codon-optimized from GeneWiz, South Plainfield, NJ, USA) coding sequences, devoid of the N-terminal signal peptide, were amplified using oligonucleotides containing in-frame 5′-BamHI and 3′-NotI (Appendix A) and cloned in a pCR8 vector, which was used as a template for the subsequent mutagenesis experiments. The LH/PLOD mutants were generated using the Phusion Site Directed Mutagenesis Kit (Thermo Fisher Scientific, Waltham, MA, USA). The entire plasmid was amplified using phosphorylated primers. For all mutants, the forward primer introduced the mutation of interest (Appendix A). The linear mutagenized plasmid was phosphorylated using T4 polynucleotide kinase (Life Technologies, Carlsbad, CA, USA) prior to ligation. All plasmids were checked via Sanger sequencing prior to cloning into the expression vector. Wild-type and mutant LH3/PLOD3 and LH1/PLOD1 mature sequences were cloned into the pUPE.106.08 mammalian expression vector (U-protein Express BV) in frame with a 6xHis-tag followed by a Tobacco Etch Virus (TEV) protease cleavage site. The GLT25D1 was sub-cloned into a modified pET28b-SUMO vector (Novagen, Paris, France), yielding the final construct bearing an N-terminal 8xHis-SUMO fusion for recombinant production using *E. coli*.

### 4.3. LH/PLOD Recombinant Expression and Protein Purification

Suspension growing HEK293-F cells (Life Technologies, Paisley, UK) were transfected at a confluence of 10^6^ cells mL^−1^ using 1 μg of plasmid DNA and 3 μg of linear polyethyleneimine (PEI; Polysciences, Warrington, PA, USA). Cells were harvested 6 days after transfection by centrifuging the medium for 15 min at 1000× *g*. The clarified medium was filtered using a 0.2 mm syringe filter, and the pH was adjusted to 8.0 prior to affinity purification as described elsewhere [33]. All proteins were isolated from the medium exploiting the affinity of the 6xHis tag for the HisTrap Excel (GE Healthcare, Chicago, IL, USA) affinity column. The purified proteins were further polished using a Superdex 200 10/300 GL (GE Healthcare) equilibrated in 25 mM HEPES/NaOH, 200 mM NaCl, pH 8.0, to obtain a homogenous protein sample; peak fractions containing the protein of interest were pooled and concentrated to 1 mg mL^−1^.

### 4.4. GLT25D1 Recombinant Expression and Protein Purification

The pET28b-SUMO-GLT25D1 plasmid was transformed in *E. coli* BL21-codonplus(DE3)-RP Plus (Agilent, Santa Clara, CA, USA), a single colony was picked and inoculated in 100 mL Luria–Bertani medium supplemented with 0.1 mg/mL Kanamycin (1:1000 *v*/*v*). This pre-culture was grown overnight at 37 °C in a shaking incubator set at 220 rpm. On the following day, this pre-culture was used to inoculate 6 L of autoinducing ZYP5052 medium [49] for large-scale production. The culture was grown for 4 h at 37 °C. Then, the culturing temperature was lowered to 20 °C for a total of 24 h prior to cell harvesting by centrifugation at 4000× *g* for 20 min. The resulting cell pellet was resuspended and homogenized in 100 mL of buffer A (25 mM HEPES/NaOH, 0.5 M NaCl, pH 8) in a 1:5 (*w*/*v*) wet cell pellet–buffer ratio and then disrupted by sonication (16 cycles, 9 s on, 6 s off pulses). The cell debris was removed via centrifugation (50,000× *g*, 40 min, 4 °C); the supernatant was then filtered through a1 μm syringe-driven filter (Minisart GF, Sartorius, Göttingen, Germany). The clarified lysate was loaded onto a His-Trap Excel 5 mL column (GE Healthcare) that was pre-equilibrated with buffer A using a NGC chromatography system (Bio-Rad, Hercules, CA, USA). Elution was carried out stepwise by adding imidazole up to a final 250 mM concentration. Fractions containing His-SUMO-GLT25D1 were pooled, subjected to imidazole removal through passage on a HiTrap Desalting 5 mL column (GE Healthcare), supplemented with 3 μg/mL of His-tagged SUMO protease (1:300 *v*/*v*), and incubated for 2 h at room temperature. The sample was then loaded onto a His-Trap Excel 5 mL column (GE Healthcare) pre-equilibrated with buffer A. Cleaved GLT25D1 eluted in the flow-through fraction. The sample was concentrated to 5 mg mL^−1^ using a 30 kDa Amicon Ultra-15 centrifugal filter concentrator (Merck, Rahway, NJ, USA). The concentrated sample was the further polished via Size Exclusion Chromatography (SEC) through the use of a Superdex 200 Increase 10/300 GL column (GE Healthcare) pre-equilibrated with SEC buffer (25 mM HEPES/NaOH, 200 mM NaCl, pH 8). Pure GLT25D1 peak fractions, as assessed by SDS-PAGE analysis, were further concentrated to 5 mg/mL, flash-frozen in liquid nitrogen, and stored at −80 °C until usage.

### 4.5. Direct Mass Spectrometry Activity Assays

Direct LH activity assays were performed using 5 μM LH/PLOD (wild-type or variants), 50 μM FeCl_2_, 100 μM 2-OG, 500 μM Ascorbate, and 1 mM peptide substrate. For Gal-T activity measurements, 5 μM GLT25D1, 50 μM Mn, 100 μM UDP-Gal were added. For Glc-T activity measurements, 100 μM UDP-Glc was also added. Each experiment was performed in at least three technical repeats. Control experiments were performed by the selective removal of either the LH/PLOD or GLT25D1 enzyme, as described in the text and in the figure captions. All reactions were allowed to proceed for 3 h at 37 °C. Moreover, 10 μL of each reaction sample were mixed with 38 μL of Milli-Q water and acidified by the addition of 2 μL of formic acid (FA) to reach a total volume of 50 μL. These samples were placed into the cooled autosampler (10 °C) of a UHPLC system (AB Sciex, Framingham, MA, USA) connected to a high-resolution QTOF mass spectrometer (AB Sciex X500B) equipped with a Turbo V Ion source and a Twin-Sprayer electrospray ionization (ESI) probe, controlled by SCIEX OS 2.1 software. Peptides were separated via reverse phase (RP) HPLC on a Hypersil Gold (Thermo Fisher Scientific) C18 column (150 × 2.1 mm, 3 μm particle size, 175 Å pore size) using a linear gradient (2–50% solvent B in 15 min), with solvent A consisting of 0.1% aqueous formic acid (FA) and solvent B acetonitrile (ACN) supplemented with 0.1% FA. Flow rate was kept constant at 0.2 mL/min. Mass spectra were collected in positive polarity under constant instrumental conditions, which were as follows: ion spray voltage 4500 V, declustering potential 100 V, curtain gas 30 psi, ion source gas 1 40 psi, ion source gas 2 45 psi, temperature 350 °C, collision energy 10 V. Spectra analyses were performed using SCIEX OS 2.1 software.

### 4.6. Indirect Luminescence-Based Activity Assays

The LH and Glc-T activities were tested using luminescence-based enzymatic assays with a GloMax Discovery (Promega, Madison, WI, USA) as described in Scietti et al. [45]. Minor modifications were established for the Glc-T competitive inhibition assays, where 1 μL of a mixture of 250 μM MnCl_2_, UDP-glucose (GlcT assay) and increasing concentrations of either UDP-GlcA or UDP-Xyl were initially added to the enzyme and gelatin substrate to start the reactions. All experiments were performed in triplicate. The control experiments were performed in the same conditions by selectively removing the LH/PLOD enzymes. Data were analyzed and plotted using GraphPad Prism 7 (Graphpad Software, San Diego, CA, USA).

### 4.7. Differential Scanning Fluorimetry (DSF)

DSF assays were performed on LH/PLOD wild-type and mutants using a Tycho NT.6 instrument (NanoTemper Technologies GmbH, München, Germany). The enzyme samples were kept at a concentration of 1 mg/mL in a buffer composed of 25 mM HEPES, 500 mM NaCl, pH 8. Binding assays were performed by incubating LH3/PLOD3 with 50 mM MnCl_2_ and 5 mM free UDP or UDP-sugar donor substrates or their analogs. Data were analyzed and plotted using GraphPad Prism 7 (Graphpad Software, USA).

### 4.8. Crystallization, Data Collection, Structure Determination, and Refinement

Wild-type and mutant LH3/PLOD3 co-crystallization experiments were performed using the hanging-drop vapor-diffusion method protocols as described elsewhere [33], i.e., by mixing 0.5 μL of enzyme concentrated at 3.5 mg mL^−1^ and, depending on the experiment performed, pre-incubating with 500 μM FeCl_2_, 500 μM MnCl_2_, and supplementing with 1 mM of the appropriate UDP-sugar analogs (UDP, UDP-glucose, UDP-glucuronic acid, UDP-xylose) with 0.5 μL of reservoir solutions composed of 600 mM sodium formate, 12% poly-glutamic-acid (PGA-LM, Molecular Dimensions, Sheffield, UK), 100 mM HEPES/NaOH, pH 7.8. Crystals were cryo-protected with the mother liquor supplemented with 20% glycerol, harvested using MicroMounts Loops (Mitegen, Ithaca, NY, USA), flash-cooled, and stored in liquid nitrogen prior to data acquisition. X-ray diffraction data were collected at various beamlines of the European Synchrotron Radiation Facility, Grenoble, France and at the Swiss Light Source, Villigen, Switzerland. Data were indexed and integrated using *XDS* [50] and scaled using *Aimless* [51]. Data collection statistics are summarized in Appendix A. The data showed strong anisotropy and therefore underwent anisotropic cut-off using *STARANISO* [52] prior to structure determination and refinement. The structures were solved by molecular replacement using the structure of wild-type LH3/PLOD3 in complex with Fe^2+^, 2-OG and Mn^2+^ (PDB ID: 6FXM) [33] as a search model using *PHASER* [53]. The structure was refined with successive steps of manual building in *COOT* [54], and automated refinement was carried out using *phenix.refine* [55]. Model validation was performed using *MolProbity* [56]. The refinement statistics for the final models are reported in Appendix A.

## Figures and Tables

**Figure 1 ijms-24-11213-f001:**
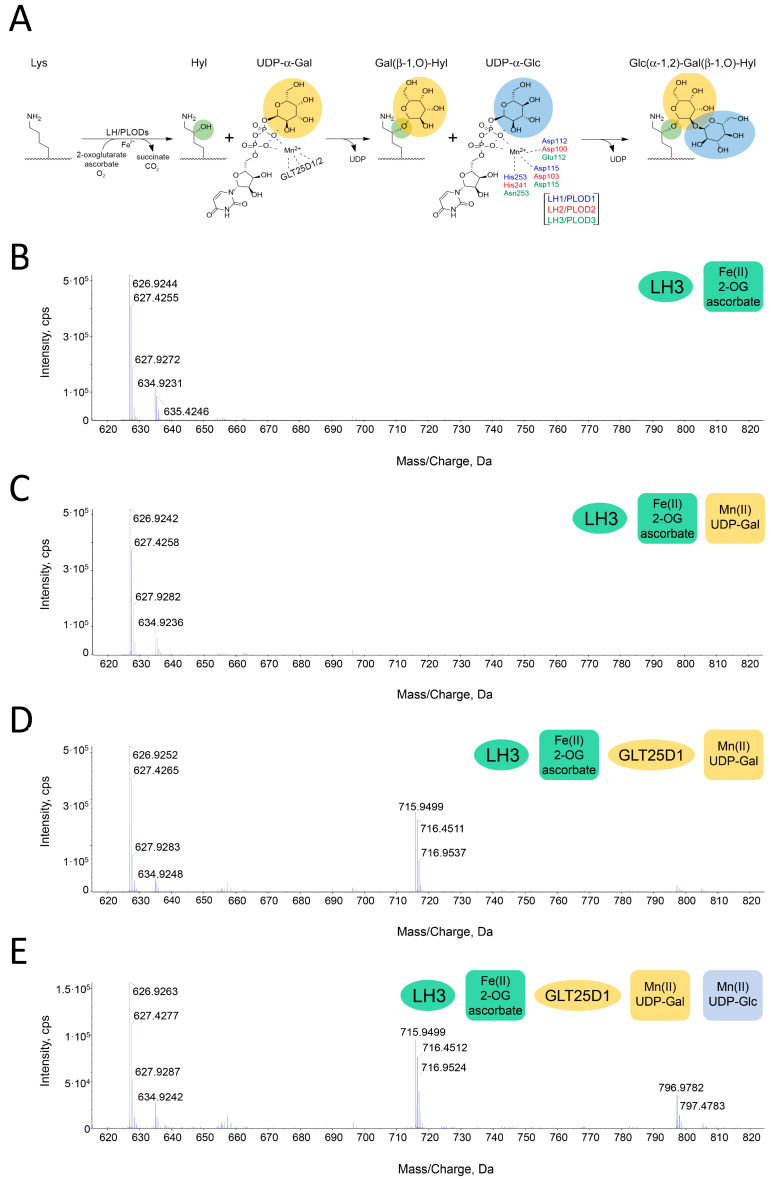
A direct assay to probe Lys-to-Glc-Gal-Hyl conversion. The reaction schematic monitored by the assay is depicted in (**A**). The MS spectra show the results obtained by incubating a synthetic GIKGIKGIKGIK peptide (MW 1254 Da) with enzymes and cofactors, as shown in the figure legends. All peaks identified are doubly charged, resulting in nominal masses corresponding to half of the expected MW. (**B**) Using LH3/PLOD3 and LH activity cofactors (i.e., 2-OG and Fe^2+^), MS peaks corresponding to a singly hydroyxylated Lys on the peptide (i.e., 635 Da) appear. (**C**) The addition of Gal-T activity cofactors (i.e., UDP-Gal and Mn^2+^) to the same mixture as in (**B**) does not yield additional MS peaks. (**D**) When the mixture in (**C**) is incubated with GLT25D1, the MS peaks corresponding to Gal-Hyl are found (i.e., 716 Da). (**E**) When the same mixture as in (**D**) also contains UDP-Glc, the peaks corresponding to Glc-Gal-Hyl appear (i.e., 797 Da).

**Figure 2 ijms-24-11213-f002:**
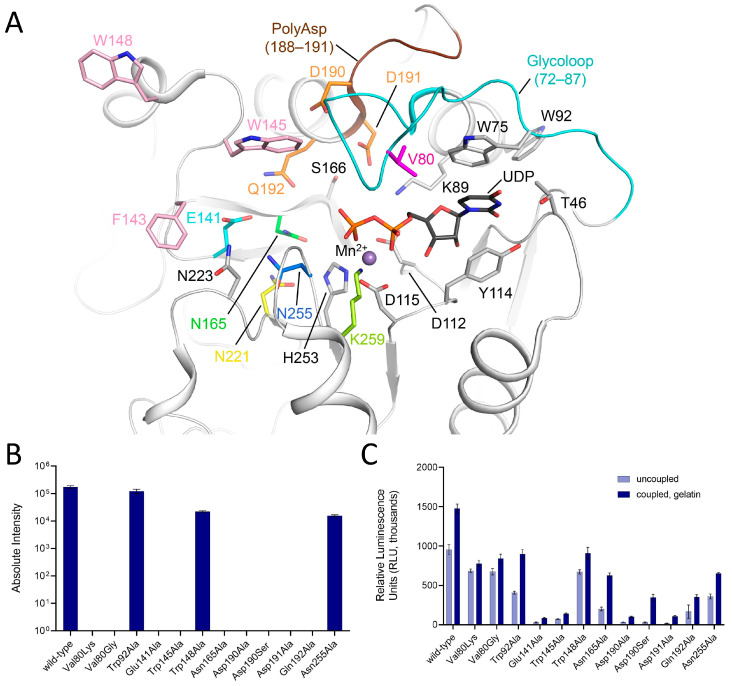
Structural and functional features of the LH3/PLOD3 glycosyltransferase (GT) domain. (**A**) Cartoon representation of the LH3/PLOD3 GT domain (PDB ID: 6FXR) showing the key residues shaping the catalytic site as sticks. The PolyAsp motif (brown) and the glycoloop (cyan) involved in the binding of UDP-sugar donor substrates are shown. The residues implicated in the catalytic activity and investigated in this works are colored, while the residues depicted in gray have already been shown to be essential in Mn^2+^ (purple sphere) and UDP (black sticks) coordination. (**B**) Summary of the evaluation of the Glc-T activity of LH3/PLOD3 mutants compared to the wild-type using MS direct assays. (**C**) Evaluation of the Glc-T activity of LH3/PLOD3 mutants compared to wild-type using luminescence-based indirect assays. Each graph shows the enzymatic activity detected in the absence (i.e., “uncoupled”, light blue) or presence of gelatin, which was used as the acceptor substrate. The plotted data are baseline-corrected, where the baseline was the background control. In both (**B**,**C**) panels, the error bars represent standard deviations from the averages of independent experiments (N > 3).

**Figure 3 ijms-24-11213-f003:**
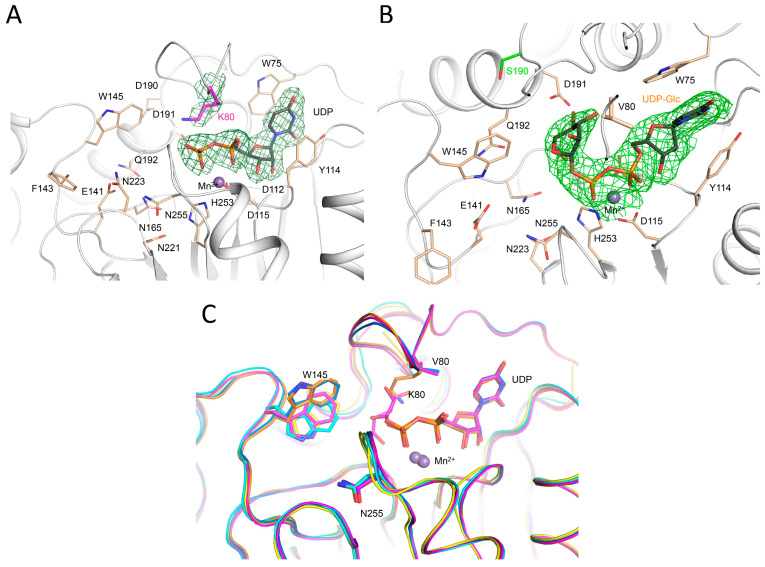
Structural characterization of LH3/PLOD3 mutants. (**A**) Crystal structure of the LH3/PLOD3 p.(Val80Lys) mutant in complex with UDP-glucose and Mn^2+^. Electron density is visible for the mutated lysine and the UDP portion of the donor substrate (green mesh, 2*F_o_*−*F_c_* omit electron density map, contoured at 1.3 σ). Catalytic residues shaping the enzyme cavity are shown as sticks; Mn^2+^ is shown as a purple sphere. Consistent with what was observed in the crystal structure of wild-type LH3/PLOD3, the glucose moiety of the donor substrate is not visible in the experimental electron density. (**B**) Crystal structure of the LH3/PLOD3 p.(Asp190Ser) mutant in complex with UDP-glucose and Mn^2+^. Electron density is visible for the mutated Serine and for the entire donor substrate, including the sugar moiety (green mesh, 2*F_o_*−*F_c_* omit electron density map, contoured at 1.3 σ). Colors and representations as in (**A**). (**C**) Superposition of wild-type, p.(Val80Lys), and p.(Asp190Ser) LH3/PLOD3 available crystal structures in substrate-free (cyan for wild-type, yellow for p.(Val80Lys), respectively) and with UDP-glucose bound (marine for wild-type, orange for p.(Val80Lys), magenta for p.(Asp190Ser), respectively) states. Notably, the conformations adopted by the side chain of Trp145 upon ligand binding are consistent in the wild-type and in the mutant enzyme. As the glycoloop is flexible in substrate-free structures, the side chains of Val/Lys80 are only visible in the in UDP-sugar-bound structures.

**Figure 4 ijms-24-11213-f004:**
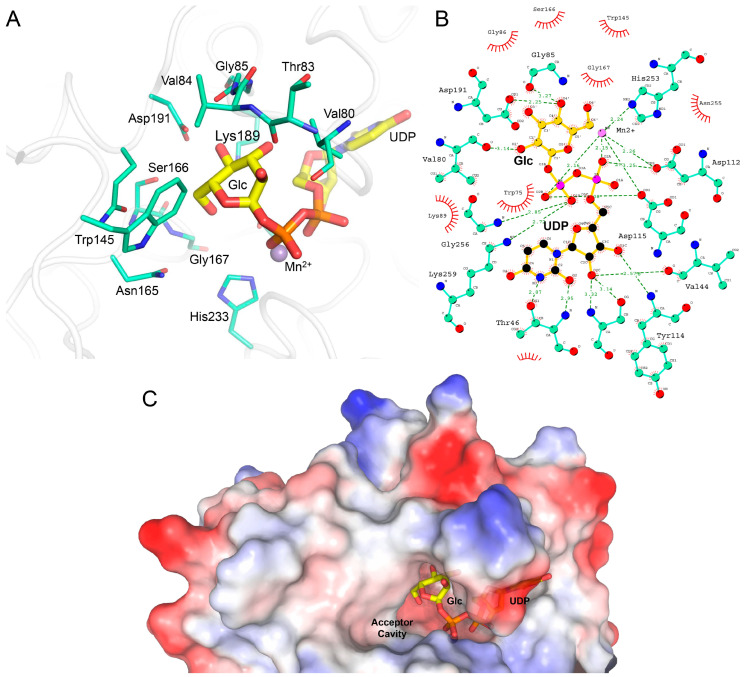
Binding mode of the glucose moiety of the UDP-Glc donor substrate observed in the crystal structure of LH3/PLOD3 p.(Asp190Ser). (**A**) Highlight of the amino acid network surrounding the Glc moiety of the donor substratein the crystal structure. UDP-Glc is shown as thick yellow sticks, whereas amino acids found at less than 5 Å distance from the Glc moiety are shown as thin blue/green sticks. (**B**) Overview of the interaction network surrounding the UDP-Glc donor substrate in the co-crystal structure with LH3/PLOD3 p.(Asp190Ser). Colors are as in (**A**). Figure made with LIGPLOT+ [36]. (**C**) The conformation adopted by the Glc moiety of the UDP-Glc substrate in the glycosyltransferase catalytic site of LH3/PLOD3 p.(Asp190Ser) leaves an empty cavity that is geometrically and sizably compatible with the Gal moiety of the acceptor substrate. Shown is a surface rendering of the GT domain of LH3/PLOD3 p.(Asp190Ser) colored by electrostatic potential, with highlights of the UDP-Glc donor substrate shown as sticks.

**Figure 5 ijms-24-11213-f005:**
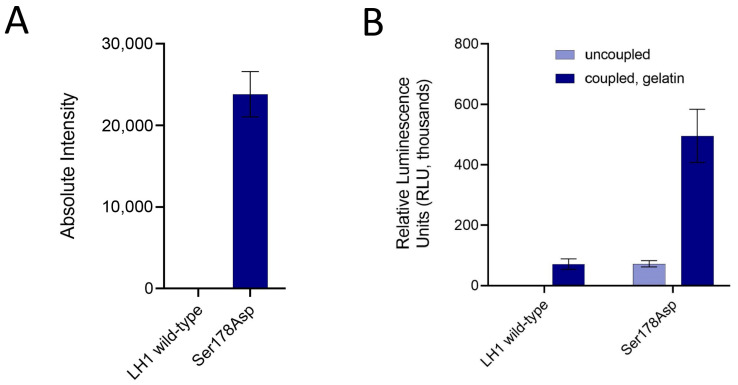
Ser178 in LH1/PLOD1, corresponding to Asp190 in LH3/PLOD3, is a key residue for Glc-T activity for both enzyme isoforms. (**A**) Direct MS-based assays comparing the signal associated with Glc-Gal-Hyl using wild-type and Ser178Asp LH1/PLOD1 variants. (**B**) Evaluation of the Glc-T activity of LH1 wild-type and Ser178Asp using luminescence-based indirect assays. The analysis of coupled and uncoupled enzymatic activities is as in Figure 2C. In both panels, the error bars represent standard deviations from the averages of independent experiments (N > 3).

**Figure 6 ijms-24-11213-f006:**
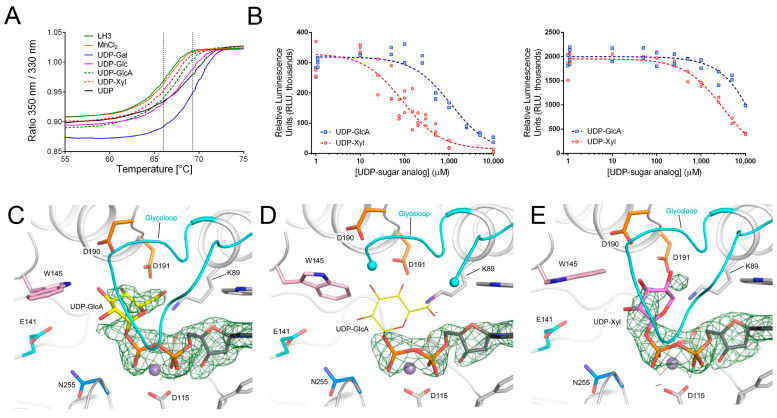
Characterization of UDP-sugar analogs. (**A**) Thermal stability of LH3 wild-type (solid green) using differential scanning fluorimetry (DSF) in the presence of various Mn^2+^ and several UDP-sugars. A prominent stabilization effect is achieved in the presence of the biological donor substrates UDP-galactose (solid blue), UDP-Glucose (solid purple), and free UDP (solid black). A milder stabilization effect is also obtained with UDP-xylose (red dash) and UDP-glucuronic acid (green dash). (**B**) Luminescence-based competition assays evaluating the binding of increasing concentrations of UDP-GlcA or UDP-Xyl to wild-type LH3/PLOD3 in the presence of either UDP-Gal (**left**) or UDP-Glc (**right**) and acceptor substrates (i.e., gelatin). (**C**) Crystal structure of LH3 wild-type in complex with Mn^2+^ and UDP-glucuronic acid shows clear electron density for UDP (2*F_o_*−*F_c_* omit electron density maps, green mesh, contour level 1.2 σ). The glucuronic acid (shown in yellow) can be modelled even if with the partial electron density. (**D**) Crystal structure of the LH3 Val80Lys mutant in complex with Mn^2+^ and UDP-glucuronic acid. While the UDP backbone can be modelled in the electron density (black sticks) (2*F_o_*−*F_c_* omit electron density maps, green mesh, contour level 1.2 σ), in this case, no electron density is present for the glucuronic acid (shown in yellow). In addition, the portion of the glycoloop containing the mutated lysine is flexible from residue 79 to 83 (shown as cyan spheres). (**E**) Crystal structure of LH3 wild-type in complex with Mn^2+^ and UDP-xylose. Similar to UDP-GlcA, UDP shows clear electron density (2*F_o_*−*F_c_* omit electron density maps, green mesh, contour level 1.2 σ), whereas partial density is shown for the xylose moiety (shown in pink).

**Table 1 ijms-24-11213-t001:** Results of activity assays (performed using either MS or luminescence) for all LH3/PLOD3 mutants described in this work compared to wild-type LH3/PLOD3.

Mutation	Localization	Folding State/ LH Activity	Glc-T Activity (MS) (%)	Glc-T Activity (Luminescence) (%)	
				Uncoupled	Coupled, Gelatin
wild-type	N/A	Yes	100	100	100
Val80Lys	glycoloop	Yes	N.D.	72 ± 2	53 ± 3
Val80Gly	glycoloop	Yes	N.D.	70 ± 4	54 ± 4
Trp92Ala	UDP-binding cavity	Yes	71 ± 14	43 ± 2	60 ± 4
Asp190Ala	poly-Asp helix	Yes	N.D.	3 ± 0.27	7 ± 0.5
Asp191Ala	poly-Asp helix	Yes	N.D.	2 ± 0.3	7 ± 0.6
Asp190Ser	poly-Asp helix	Yes	N.D.	2 ± 0.4	17 ± 2
Trp145Ala	acceptor substrate cavity and gates	Yes	N.D.	8 ± 0.5	9 ± 0.6
Trp148Ala	acceptor substrate cavity and gates	Yes	13 ± 2	70 ± 3	61 ± 5
Asn165Ala	region proximate UDP-sugar	Yes	N.D.	21 ± 3	42 ± 2
Gln192Ala	region proximate UDP-sugar	Yes	N.D.	17 ± 9	24 ± 2
Glu141Ala	region proximate UDP-sugar	Yes	N.D.	3 ± 0.6	6 ± 0.6
Asn255Ala	region proximate UDP-sugar	Yes	9 ± 0.5	38 ± 3	44 ± 0.7
Pro270Leu	interface of AC and GT domains	No	N.D.	N.D.	N.D.

**Table 2 ijms-24-11213-t002:** Evaluation of the competitive binding of UDP-GlcA and UDP-Xyl in wild-type LH3/PLOD3 in the presence of UDP-Gal or UDP-Glc and acceptor substrates.

Inhibitor	UDP-Gal (IC_50_, μM)	UDP-Glc (IC_50_, μM)
wild-type + UDP-GlcA	1130 ± 370	>10,000
wild-type + UDP-Xyl	91 ± 23	3170 ± 211

## Data Availability

Coordinates and Structure Factors of the new molecular structures presented in this study have been deposited in the Protein Data Bank under accession codes 6TE3, 6TES, 6TEC, 6TEU, 6TEX, 6TEZ, 8ONE. Other data are available from the corresponding author upon reasonable request.

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
