# Peer review of "Identification of Regulatory Molecular “Hot Spots” for LH/PLOD Collagen Glycosyltransferase Activity"

_ijms, 2023, doi:10.3390/ijms241311213_

Round 1

Reviewer 1 Report

This group’s previous LH3 structural work (Scietti et al., 2018) is very interesting. This manuscript is the follow-up. The valuable data in this paper include the validation of LH3’s collagen glucosyltransferase activity. The structure of LH3D190S-UDP glucose is also novel, however, the structure was poorly supported by the experimental data. Since these authors have already established a UDP/UDP-glucose binding assay, it will be very helpful if these authors can make the relevant mutations based on LH3-UDP glucose structural model and confirm the contribution of these residues in binding the sugar moiety of UDP glucose, instead of generating irrelevant mutations such as Trp148Ala. These findings will make a good story. The extra structural data in the manuscript are mostly negative results and add little to the previously published work (Scietti et al., 2018) due to the poor electron density near the active site.

Here are the detailed comments:

These authors crystallized LH3 with different types of UDP-sugars other than UDP-glucose. However, no clear density was observed for the sugar moiety. 

Since LH1 “glycoloop” is one amino acid shorter than that of LH3, the sequence alignment between LH1 and LH3 needs to be carefully performed. Depending on how the gap is handled, LH3 Val80 can be aligned with LH1 Gly69 or Lys68 residue. More importantly, LH1 Gly69 and LH2 Gly80 align well if we only compare LH1 and LH2. Based on these findings, it looks like LH3 Val80 is replaced with Gly in both LH1 and LH2.  So, the rationale of LH3 Val80Lys mutation has no basis. Introducing a long-chain amino acid in the wrong place could force the UDP sugar to bind LH3 in a biologically irrelevant way. 

The assignment of the electron density blob to Lys80 sidechain is questionable. If the main chain is not visible, how do we know the blob is Lys80 sidechain? 

These authors suggested a long-range interaction between Trp75 and Trp92; however, no experimental data were provided to support this claim. 

Glu141 is located at the end of a beta strand >10 angstroms away from the Mn2+. How could Glu141 be involved in catalysis based on the current structural models?

“Surprisingly, the increase in thermal stability did not correlate with efficient trapping of the reaction product in wild-type LH3/PLOD3 molecular structures.” How do we know the product is not there? It may be just invisible due to flexibility. 

Besides these author work, Guo et al., 2021 also suggested LH1/PLOD1 has Glc-T activity.

The paper is clearly written.

Author Response

We would like to thank the reviewers for the time spent going through our work, highlighting its strengths and weaknesses, and providing useful suggestions for revision. We duly appreciated that, by combining the feedbacks received by the two reviewers, our combined structural and biochemical work has been appreciated and shows merit suitable for publication in your Journal. We have now revised our text incorporating the reviewer’s suggestions, also adding experimental results about a new LH3/PLOD3 mutant (Val80Gly) that we recently managed to complete. The following sections include a detailed point-by-point response to reviewer’s comments.

REVIEWER 1

This group’s previous LH3 structural work (Scietti et al., 2018) is very interesting. This manuscript is the follow-up. The valuable data in this paper include the validation of LH3’s collagen glucosyltransferase activity. The structure of LH3D190S-UDP glucose is also novel, however, the structure was poorly supported by the experimental data. Since these authors have already established a UDP/UDP-glucose binding assay, it will be very helpful if these authors can make the relevant mutations based on LH3-UDP glucose structural model and confirm the contribution of these residues in binding the sugar moiety of UDP glucose, instead of generating irrelevant mutations such as Trp148Ala. These findings will make a good story. 

We thank the reviewer for his/her comments. Actually, we wish to point out that the current work includes the structural investigation of LH3/PLOD3 Asp190Ser mutant as part of a more comprehensive investigation of the GT domain, its features regarding molecular recognition of UDP-Glc as well as other non-substrate molecules (such as UDP-GlcA and UDP-Xyl), plus the first direct in vitro detection of glucosylgalactosyltransferase activity in multifunctional human lysyl hydroxylase enzymes. The mutagenesis screening performed indeed includes several residues observed in the surrounding of the Glc moiety, such as Trp145, Asn165, Glu191, Gln192 and a thorough evaluation of the significance of these residues to catalysis is provided by extensive evaluation of the catalytic activity of each mutant using two orthogonal enzymatic assays. The inclusion of Trp148Ala mutation was based on the hypothesis that, given the extended nature of the collagen polypeptide acceptor substrate processed by LH/PLOD enzymes, the presence of a hydrophobic residue on the enzyme’s surface adopting distinct conformations in presence or absence of donor substrate could provide some indications regarding the conformation adopted by the LH/PLOD GT domain to facilitate collagen binding. Evaluating experimentally such possibility does not seem irrelevant to us.

The extra structural data in the manuscript are mostly negative results and add little to the previously published work (Scietti et al., 2018) due to the poor electron density near the active site.

We respectfully disagree with the reviewer about this point, considering that the structural results obtained, for both wild-type and mutant LH3/PLOD3 reveal that UDP-GlcA and UDP-Xyl indeed bind to the GT pocket positioning the UDP moiety of the molecules in a nearly identical fashion as UDP-Glc substrate, strengthening the indication for a promiscuous binding site for several UDP-sugar molecules and their analogs for inhibitor development. In this respect, the negative result regarding absence of UDP bound in the LH3/PLOD3 structure is intriguing and we think that the reasoning proposed in the discussion section of our manuscript, proposing a release mechanism for UDP immediately after donor substrate processing (and regardless of actual transfer of the sugar moiety to a possible acceptor substrate), is relevant. We have revised some statements in our results and discussion sections to further clarify our observations (lines 601-607): “The absence of experimental density in co-crystal structures of LH3/PLOD3 with free UDP indicates that retention of donor substrates in the GT domain of LH/PLODs requires the presence of both the UDP and the sugar moiety and that, regardless of the presence of glycan acceptor substrates, after hydrolysis free UDP is rapidly released from the catalytic site, possibly through conformational changes involving the flexible glycoloop. In this respect, the new data obtained for LH3/PLOD3 in complex with UDP-sugar analogs acting as mild inhibitors (Figure 5)

These authors crystallized LH3 with different types of UDP-sugars other than UDP-glucose. However, no clear density was observed for the sugar moiety. 

We agree that the density for the sugar moiety of the UDP analogs is not providing a single, unambiguous conformation for the sugar, and in fact we stated in our manuscript that we refrain from providing an accurate analysis of the possible interactions between residues of the GT domain and the sugar moiety because of the flexibility associated to the latter. Nevertheless, the weak density associated to these molecules corroborates the biochemical data associated to their weak inhibition of glycosyltransferase activity, further supporting the notion that the sugar binding pocket for the donor substrate can host multiple sugar moieties (isosteric to glucose but with different stereochemistry such as galactose; smaller than glucose, like xylose; as well as larger, like glucuronic acid) and that acting on the individual substituents surrounding the sugar ring can be a valid strategy for inhibitor development.

Since LH1 “glycoloop” is one amino acid shorter than that of LH3, the sequence alignment between LH1 and LH3 needs to be carefully performed. Depending on how the gap is handled, LH3 Val80 can be aligned with LH1 Gly69 or Lys68 residue. More importantly, LH1 Gly69 and LH2 Gly80 align well if we only compare LH1 and LH2. Based on these findings, it looks like LH3 Val80 is replaced with Gly in both LH1 and LH2. So, the rationale of LH3 Val80Lys mutation has no basis. Introducing a long-chain amino acid in the wrong place could force the UDP sugar to bind LH3 in a biologically irrelevant way. 

Prior to submitting our manuscript, we have thoroughly checked sequence alignments as well as computational models for all three human LH/PLOD isoforms. Our rationale was indeed to obtain reliable in silico comparisons prior to performing experimental investigations. We performed several sequence alignments considering the variancy related to one less amino acid in LH1/PLOD1 glycoloop compared to other isoforms. The reviewer states that “LH1/PLOD1 Gly69 and LH2/PLOD2 Gly80 align well if we only compare LH1/PLOD1 and LH2/PLOD2”, but (1) such alignment leaves a gap immediately afterwards and misaligns the negatively charged residue immediately preceding the Gly residue subject of discussion, which is conserved in all three isoforms and (2) having at hand multiple sequences, the larger dataset should be handled to prevent positioning bias. In this respect, we took into consideration both possibilities: the presence of a Lys residue, as in LH1, and the presence of a Gly residue, as in LH2/PLOD2 (now added in the revised version of our manuscript). In addition, we consider the structural data obtained for the LH3/PLOD3 Val80Lys mutant nevertheless useful for inhibitor design, as they highlight that the presence of a long-chain amino acid in such position is compatible with donor substrate binding in the cavity.

The assignment of the electron density blob to Lys80 sidechain is questionable. If the main chain is not visible, how do we know the blob is Lys80 sidechain? 

The glycoloop is flexible, hence not visible in electron density, when no substrate/inhibitor is bound. With UDP-Glc co-crystallized, the whole glycoloop is visible and the electron density is clearly visible. We invite the reviewer to check the density shown on figure panel 3A, and if not yet convinced, to check our deposited dataset with PDB ID 6TEX.

These authors suggested a long-range interaction between Trp75 and Trp92; however, no experimental data were provided to support this claim. 

We have revised the statement regarding the positioning of these two residues to clarify our observations (lines 332-334): “These findings suggest that the presence of non-conserved residues within the LH3/PLOD3 GT domain distant from those involved in first-shell interactions with donor and acceptor substrates may contribute to the productive conformations of the glycoloop in donor substrate-bound states.

Glu141 is located at the end of a beta strand >10 angstroms away from the Mn2+. How could Glu141 be involved in catalysis based on the current structural models?

We thank the reviewer for suggesting us to further improve the description of this important site. We have revised the text in the result section and added our reasoning at lines 428-434: “Conversely, the p.(Glu141Ala) mutant was completely inactive, incapable of UDP-donor substrate activation (Figure 2B-C, Table 1). These suggest essential roles for Glu141 in catalysis, either through initial binding of UDP-Glc donor substrates prior to their final positioning in the catalytic site or through stabilization of water molecule networks in the large GT cavity enabling donor substrate processing. The surrounding negatively charged pocket composed of Asp190, Asp191, Gln192, Asn165, all residues found relevant, but not essential for catalysis, likelyassists the glycosyltransferase activity.

“Surprisingly, the increase in thermal stability did not correlate with efficient trapping of the reaction product in wild-type LH3/PLOD3 molecular structures.” How do we know the product is not there? It may be just invisible due to flexibility. 

Given the consistent results observed for all UDP-sugar bound structures showing that UDP adopts the exact same conformation regardless the sugar moiety (RMSD all atoms for the UDP region is less than 0.2 angstrom when superimposing all deposited LH3/PLOD3 structures with bound substrates, analogs or inhibitors), the absence of density for UDP is unlikely to correlate with a bound substrate showing pronounced flexibility. In our view, it is rather logic to hypothesize instead that UDP binding is transient and that UDP is not trapped in the crystal structure.

Besides these author work, Guo et al., 2021 also suggested LH1/PLOD1 has Glc-T activity.

We agree with the reviewer that both Guo et al., 2021 and that Koenig et al., 2021 have suggested the intriguing possibility of Glc-T activity for LH1/PLOD1. However, none of the previous publications managed to provide experimental evidence of glucose attachment to galactosylhydroyxlysine residues, as both papers made use of indirect UDP-Glo assays to characterize the putative Glc-T activity. In this respect, it should be noted that using the same assay it is impossible to discriminate between spurious processing of UDP-Gal into UDP + Gal and actual galactosyltransferase for LH3/PLOD3. Thus, we think that in our text we have duly cited the previous work done by us (Koenig et al., 2022) and others (Guo et al., 2021).

Reviewer 2 Report

In their manuscript, Mattoteia and colleagues report the role of two enzymes, multifunctional collagen lysyl hydroxylase 3 (LH3/PLOD3) and collagen galactosyltransferase GLT25D1, in hydroxylysine glycosylations. Employing indirect luminescence tests, direct mass spectrometry assays, and molecular structure analyses, they demonstrate that LH3/PLOD3 has only Glc-T activity, whereas GLT25D1 has only Gal-T activity. These findings elucidate the intricate network of geometries, charges, and interactions that enable LH3/PLOD3 glycosyltransferase activities. The content of this paper is fluent and fully demonstrated. Overall, I suggest this manuscript be published in the International Journal of Molecular Sciences.

 I hope the following suggestions will help to revise this article.:

 Major:

 Although Mn2+ ions are generally crucial in the enzyme-catalyzed reaction process, the manuscript provides limited information on their specific roles. It is recommended to append more discussion about Mn2+ ions, such as a diagram similar to the reaction path.

Minor:

 1. Figure 3 appears in front of figure 2, which seems a little strange.

2. Table 1 is divided into A and B, which is not a common practice. It is recommended to split it into tables 1 and 2. In addition, the font and size of Table caption are not the same.

Author Response

We would like to thank the reviewers for the time spent going through our work, highlighting its strengths and weaknesses, and providing useful suggestions for revision. We duly appreciated that, by combining the feedbacks received by the two reviewers, our combined structural and biochemical work has been appreciated and shows merit suitable for publication in your Journal. We have now revised our text incorporating the reviewer’s suggestions, also adding experimental results about a new LH3/PLOD3 mutant (Val80Gly) that we recently managed to complete. The following sections include a detailed point-by-point response to reviewer’s comments.

REVIEWER 2

In their manuscript, Mattoteia and colleagues report the role of two enzymes, multifunctional collagen lysyl hydroxylase 3 (LH3/PLOD3) and collagen galactosyltransferase GLT25D1, in hydroxylysine glycosylations. Employing indirect luminescence tests, direct mass spectrometry assays, and molecular structure analyses, they demonstrate that LH3/PLOD3 has only Glc-T activity, whereas GLT25D1 has only Gal-T activity. These findings elucidate the intricate network of geometries, charges, and interactions that enable LH3/PLOD3 glycosyltransferase activities. The content of this paper is fluent and fully demonstrated. Overall, I suggest this manuscript be published in the International Journal of Molecular Sciences.

We sincerely thank the reviewer for his/her summary analysis of our work.

 I hope the following suggestions will help to revise this article.:

Major:

 Although Mn2+ ions are generally crucial in the enzyme-catalyzed reaction process, the manuscript provides limited information on their specific roles. It is recommended to append more discussion about Mn2+ ions, such as a diagram similar to the reaction path.

Following reviewer’s suggestion, we have better specified the Mn2+-dependent mechanisms of glycosyltransferase catalysis in the introduction, and modified Figure 1 introducing a diagram illustrating the reaction catalyzed by LH/PLOD and GLT25D1.

Minor:

  1. Figure 3 appears in front of figure 2, which seems a little strange.

We have double-checked our text, but could not find any swaps in the order of figures. Figure 3 is first mentioned in line 304, whereas Figure 2 comes earlier (line 278). Is the reviewer referring to supplementary figure 3 appearing at line 271? If so, supplementary figure 2 is mentioned at line 250 and supplementary figure 1 at line 248.

  1. Table 1 is divided into A and B, which is not a common practice. It is recommended to split it into tables 1 and 2. In addition, the font and size of Table caption are not the same.

We have adjusted the tables according to reviewer’s suggestions.

Round 2

Reviewer 1 Report

The revised manuscript improved somewhat, but some major concerns remain unresolved.

  1. For LH3D190S-UDP-glucose structure, the revised manuscript did not provide any new insights, such as the residues that directly form hydrogen bonds with glucose and the basis of LH3 sugar specificity. No direct UDP-glucose binding assay was performed. So, the structural model is not experimentally validated.
  2. The authors raised an interesting point that W148 might be involved in collagen binding. A collagen binding assay will be helpful to test this possibility.
  3. The rapid UDP release mechanism proposed here is interesting. However, it needs to be approved by comparing LH3’s UDP-glucose- and UDP- binding affinities. UDP/LH3 cocrystallization experiment resulting in no UDP density could be due to UDP’s low on rate, thus it does not provide direct support of the hypothesis.
  4. The difference between coupled vs. uncoupled enzymatic activities may be due to collagen-binding defects. For example, Val80 may be involved in collagen binding. So, the alternative possibility of the following statement should be mentioned.

“Val80 might be involved in the productive positioning of the donor substrate during transfer 299 of the glycan moiety to the acceptor substrate...”

5.     The structures in Figure 5 did not add much but raised questions. These authors claimed these sugar moieties positioned similarly in the active site as glucose, and LH3 can host multiple sugar moieties. If this is the case, why LH3 does not transfer all sugar types to collagen? Thus, it is possible that these findings may be due to two possibilities: 1, the sugar conformations in Figure 5 are not accurate. 2. The D190S mutation may force the UDP-glucose to adopt a non-productive conformation, raising concerns about the relevance of this structure. 

6.     In sum, it will be helpful to perform vigorous protein biochemistry experiments to fully validate the LH3D190S-UDP-glucose structure. So far, it is not clear how would the unvalidated sugar positioning in LH3D190S-UDP-glucose and fussy densities in figure 5 would provide a valid strategy for inhibitor design as claimed by these authors.

7.     It looks like that all LH3 crystals were obtained using the same condition. Please confirm.

Author Response

  1. For LH3D190S-UDP-glucose structure, the revised manuscript did not provide any new insights, such as the residues that directly form hydrogen bonds with glucose and the basis of LH3 sugar specificity. No direct UDP-glucose binding assay was performed. So, the structural model is not experimentally validated.

We have performed additional UDP-glucose binding testing using DSF on LH3/PLOD Asp90Ser, and compared the results obtained with those of wild-type LH3/PLOD3. This has been included in lines 351-352 “without affecting the ability of this mutant to bind UDP-Glc donor substrates (Suppl. Fig. 4E)”, and in supplementary figure 4E. We wish to point out that, stimulated by the reviewer’s suggestion, we attempted performing a series of experiments using microscale thermophoresis and isothermal titration calorimetry to possibly quantify the binding affinity of UDP-Glc to LH3/PLOD3 wild-type and mutant, but due to experimental issues associated to the marked labeling recalcitrance of LH3/PLOD3 for MST and the extensive sample requirements of ITC, we did not obtain results that we consider reliable for publication. We think that the qualitative DSF analysis presented is sufficient to corroborate what presented in the LH3/PLOD3 D190S crystal structure data, i.e., UDP-Glc binding matching what expected for LH3/PLOD3 wild-type.

We have also added more detailed description of the amino acid network surrounding the observed UDP-Glc donor substrate at lines 358-363 “The side chains of Asp191, Lys89 and Ser166, as well as the main chain carbonyl of Thr83 were found at distances and orientations possibly compatible with electrostatic contacts with the Glc moiety of the donor substrate, resulting in trapping of the sugar in a portion of the catalytic cavity, leaving a large pocket shaped by residues Glu141, Phe143, Asn165, Asn223, Asn255 available to likely host the Gal moiety of the acceptor substrate for catalysis (Figure 4)(now also shown with details in the new figure 4), with a summary description in the discussion at lines 555-556, “, enabling identification of the amino acid residues directly involved in Glc binding and those shaping the portion of the cavity likely hosting the Gal moiety of the acceptor substrate”.

  1. The authors raised an interesting point that W148 might be involved in collagen binding. A collagen binding assay will be helpful to test this possibility.

We agree with the reviewer, however, considering that the entire work that we are currently presenting is focused on the GT domain catalytic site and the impact of mutations mostly on donor substrates, we feel that such analysis (that requires extensive work associated to new fresh batches of recombinant wild-type and mutant LH3, followed by development of collagen binding experiments) is beyond the purpose of this work and, without extensive additional investigations on the acceptor substrate binding site outside the catalytic site would not provide anything more conclusive than the currently proposed hypothesis. In this respect, we prefer to leave our speculation on Trp148 as is in the manuscript. We have slightly modified the text W148, to emphasize that this is currently a speculative hypothesis.

The rapid UDP release mechanism proposed here is interesting. However, it needs to be approved by comparing LH3’s UDP-glucose- and UDP- binding affinities. UDP/LH3 cocrystallization experiment resulting in no UDP density could be due to UDP’s low on rate, thus it does not provide direct support of the hypothesis.

We have extensively compared UDP-sugar versus free UDP binding using DSF, but as previously stated we could not perform quantitative measurements on UDP and UDP-Glc binding affinities. In our opinion, the reviewer’s hypothesis on low on rates for UDP binding kinetics is unlikely given the DSF results obtained without extensive small molecule incubation prior to the 3-minute experiment performed using the Tycho instrument. In this respect, we think that the present version of our manuscript is offering a solid hypothesis to our readers based on available data, without making exaggerated claims regarding possible molecular mechanisms. No further changes made regarding this point.

The difference between coupled vs. uncoupled enzymatic activities may be due to collagen-binding defects. For example, Val80 may be involved in collagen binding. So, the alternative possibility of the following statement should be mentioned.

“Val80 might be involved in the productive positioning of the donor substrate during transfer 299 of the glycan moiety to the acceptor substrate...”

We thank the reviewer for this suggestion. Such alternative possibility has been included in line 539-541.

  1. The structures in Figure 5 did not add much but raised questions. These authors claimed these sugar moieties positioned similarly in the active site as glucose, and LH3 can host multiple sugar moieties. If this is the case, why LH3 does not transfer all sugar types to collagen? Thus, it is possible that these findings may be due to two possibilities: 1, the sugar conformations in Figure 5 are not accurate. 2. The D190S mutation may force the UDP-glucose to adopt a non-productive conformation, raising concerns about the relevance of this structure. 

We wish to point out that nowhere in our manuscript we stated that the sugar moieties of UDP-GlcA or UDP-Xyl “positioned similarly in the active site as glucose”. What we stated is that these two inhibitors do bind into the catalytic cavity showing UDP moieties perfectly matching with that found when using UDP-Glc, and that the sugar moieties “could not be interpreted with a single inhibitor conformation” (line 485). We never claimed that we have unambiguously modeled the GlcA nor the Xyl moiety in the catalytic site with perfect identification of the position of each atom, as the experimental electron density does not allow to do this. Nevertheless, the difference between substrate and inhibitor is evident and remarkable from a structural point of view: whilst co-crystal structures with donor substrate displays no density for the sugar moiety because it has been hydrolized due to enzymatic activity, or very low occupancy for a residual attached flexible moiety that shows no density at all, the inhibitor shows density for the glycan part connected to UDP because hydrolysis is not taking place. This scenario is very similar to what we observed for LH3/PLOD3 Asp190Ser mutant, where the significant reduction in catalytic activity (with no alterations in UDP-Glc binding affinity) results in a visible Glc moiety in the catalytic cavity. We have added such reasoning to our discussion at lines 606-613.

  1. In sum, it will be helpful to perform vigorous protein biochemistry experiments to fully validate the LH3D190S-UDP-glucose structure. So far, it is not clear how would the unvalidated sugar positioning in LH3D190S-UDP-glucose and fussy densities in figure 5 would provide a valid strategy for inhibitor design as claimed by these authors.

We are convinced that the results shown in this manuscript are the sole result of rigorous biochemistry and we insist on the fact that, as stated in our previous reply to this reviewer, the present work goes beyond the report of the LH3 D190S structure and the associated binding mode of UDP-Glc inside the catalytic cavity. The present revisions add further new biochemical data addressing most reviewer’s requests, plus extensive statements and graphics to highlight the features found in the new structure. Inhibitor design depends on structural mapping, and the present work provides the most in-depth structure-driven mapping of the glycosyltransferase catalytic site of a human LH enzyme. Knowing that glycan moieties can adopt bent conformations, localizing near Asp191 rather than Asn165, and that such conformational arrangements are compatible with both substrates and inhibitors, is clearly something that supports future inhibitor design aimed at filling the cavity through possible extension of glycan-like moieties that could interfere with acceptor substrate binding. This is a strategy that we (and others as well (personal communication), based on the data from this manuscript previously presented in the pre-print version) are currently exploring to design Glc-T inhibitors.

  1. It looks like that all LH3 crystals were obtained using the same condition. Please confirm.

We confirm that what we have stated in the materials and methods section regarding LH3 crystallization conditions is correct. The most successful experiments yielding the structural data presented in this work were performed using an optimized cocktail that enabled crystallization of LH3 wild-type and mutants, as well as their complexes with metal ions and donor substrates. We have tested various optimization strategies, but the condition identified systematically resulted in the best-diffracting crystals and electron density maps.

Round 3

Reviewer 1 Report

Some statements can be improved further. For example, these authors mentioned that collagen substrate is out of the scope of this work, however, it will be better if these authors can consider the collagen substrate when explaining the results. For example, the uncoupled activity does involve collagen substrate. So, how do we know the changes in V80K uncoupled activity are due to donor substrate positioning, not collagen binding? 

294 Consistently, this residue might be involved in

295 the productive positioning of the donor substrate during transfer of the glycan moiety to the acceptor

296 substrate, rather than stabilizing the UDP moiety in the catalytic pocket.

Author Response

Some statements can be improved further. For example, these authors mentioned that collagen substrate is out of the scope of this work, however, it will be better if these authors can consider the collagen substrate when explaining the results. For example, the uncoupled activity does involve collagen substrate. So, how do we know the changes in V80K uncoupled activity are due to donor substrate positioning, not collagen binding? 

294 Consistently, this residue might be involved in

295 the productive positioning of the donor substrate during transfer of the glycan moiety to the acceptor

296 substrate, rather than stabilizing the UDP moiety in the catalytic pocket.

This has been addressed precisely by the sentence reported at lines 294-296, as uncoupled activity is measured in absence of acceptor (i.e., collagen) substrate. The absence of acceptor substrate is explicitly stated in lines 289-291: “[…] we investigated the impact of the p.(Val80Lys) mutation in both absence (uncoupled activity) and presence (coupled activity) of acceptor substrates by performing luminescence-based assays”. Nevertheless, to improve readability and avoid possible confusion throughout the manuscript, we have slightly modified few sentences in both results and discussion sections.